# Furan as Impurity in Green Ethylene and Its Effects on the Productivity of Random Ethylene–Propylene Copolymer Synthesis and Its Thermal and Mechanical Properties

**DOI:** 10.3390/polym15102264

**Published:** 2023-05-11

**Authors:** Joaquín Hernández-Fernández, Esneyder Puello-Polo, Edgar Márquez

**Affiliations:** 1Chemistry Program, Department of Natural and Exact Sciences, San Pablo Campus, University of Cartagena, Cartagena 130015, Colombia; 2Chemical Engineering Program, School of Engineering, Universidad Tecnológica de Bolivar, Parque Industrial y Tecnológico Carlos Vélez Pombo Km 1 Vía Turbaco, Cartagena 130001, Colombia; 3Department of Natural and Exact Science, Universidad de la Costa, Barranquilla 080002, Colombia; 4Group de Investigación en Oxi/Hidrotratamiento Catalítico Y Nuevos Materiales, Programa de Química-Ciencias Básicas, Universidad del Atlántico, Puerto Colombia 081001, Colombia; 5Grupo de Investigaciones en Química Y Biología, Departamento de Química Y Biología, Facultad de Ciencias Básicas, Universidad del Norte, Carrera 51B, Km 5, Vía Puerto Colombia, Barranquilla 081007, Colombia

**Keywords:** furan, green ethylene, Ziegler–Natta catalyst, random copolymer, catalyst, mechanical properties, melt flow index

## Abstract

The presence of impurities such as H_2_S, thiols, ketones, and permanent gases in propylene of fossil origin and their use in the polypropylene production process affect the efficiency of the synthesis and the mechanical properties of the polymer and generate millions of losses worldwide. This creates an urgent need to know the families of inhibitors and their concentration levels. This article uses ethylene green to synthesize an ethylene–propylene copolymer. It describes the impact of trace impurities of furan in ethylene green and how this furan influences the loss of properties such as thermal and mechanical properties of the random copolymer. For the development of the investigation, 12 runs were carried out, each in triplicate. The results show an evident influence of furan on the productivity of the Ziegler–Natta catalyst (ZN); productivity losses of 10, 20, and 41% were obtained for the copolymers synthesized with ethylene rich in 6, 12, and 25 ppm of furan, respectively. PP0 (without furan) did not present losses. Likewise, as the concentration of furan increased, it was observed that the melt flow index (MFI), thermal (TGA), and mechanical properties (tensile, bending, and impact) decreased significantly. Therefore, it can be affirmed that furan should be a substance to be controlled in the purification processes of green ethylene.

## 1. Introduction

Currently, the chemical, petrochemical, and agri-food industries and the research community in general aim to mitigate the environmental impact these sectors have generated over the years [1]. For this reason, one of the strategies is to optimize the production processes of polymeric substances of fossil origin that function as raw materials to obtain products of great commercial utility [2,3]. However, the poor disposal of fossil substances and their derivatives causes damage to the health and environment of the communities near the plants that produce these substances [1,2,3,4]. This has led to the implementation of new ideas, such as using raw materials that allow the production of polymers through processes free of fossil fuels [1,2,3,4,5,6,7].

For the industrial synthesis of polymers, it is common to use the raw material ethylene from fossil fuels and endothermic processes [1,2,3,4,5,6,7,8,9]. The green ethylene obtained from the fermentation of corn, glucose, and starch has been implemented to meet the global demand for ethylene of fossil origin and thus reduce the environmental impact [1]. One route for obtaining green ethylene is from the pyrolysis of methanol [9]. Regardless of the ethylene and propylene source, these hydrocarbons are used to synthesize random or impact copolymers since they are of great academic and industrial interest for better physical and mechanical properties [10,11,12,13,14]. One of the limitations of ethylene of fossil origin is the presence of impurities of sulfur-containing chemical compounds, ketones, alcohols, arsine, phosphine, carboxylic acids, and thiols that act as aggressive inhibitors of the Ziegler–Natta catalytic systems [2], which act as a catalytic agent in the synthesis of copolymers. It is necessary to highlight that these impurities can affect other properties of the polymer such as the melt flow index (MFI), thermal degradation, molecular weight, and mechanical properties [2], which affect the productivity of the catalysts and the physicochemical properties of the synthesized copolymers. Since green ethylene is of natural origin, it is understood that it is not in the presence of these contaminants, thus having another competitive advantage. Green ethylene may have another type of contaminant present in trace concentrations since it is obtained from the transformation of natural products and methanol. In this research, we focus on heterocyclic compounds such as furan. Furan is the name by which a five-membered aromatic heterocyclic compound is recognized [14]; it is soluble in organic media, transparent and colorless with a high degree of flammability, and very volatile [14,15]. The constant study of the structure and properties of this compound has allowed us to affirm that around 135 different isomers of it are known to date [16], and many of them generate negative impacts both on health and on the production costs of certain substances and raw materials in the industrial sector. The slightest impurities in a raw material can affect the final stages of the production of copolymers [1]. In the identification of furan in renewable and non-renewable sources, techniques such as infrared radiation [1,11] and gas chromatography [2] have been used. To date, very little information is available on how compounds such as furan can influence the copolymer, how it affects the physicochemical and mechanical properties of this product, and how it can affect its production rate.

Since furan can be present as an impurity in green ethylene and can also be formed during the pyrolysis of natural waste, we will develop this research and propose reaction mechanism pathways that explain furan formation, the reaction of furan with the Ziegler–Natta catalyst, and the effect of different concentrations of furan on the efficiency of copolymer synthesis, and we will use other instrumental techniques to determine how the melt index, molecular weight distribution, thermal stability, and mechanical properties of the copolymer are affected.

## 2. Materials and Methods

### 2.1. Polymerization Process: Synthesis of Random Ethylene–Propylene Copolymer

Table 1 shows the list of reagents used to obtain polypropylene. For the development of this research, a fourth-generation spherical Ziegler–Natta catalyst was used, with MgCl_2_ support with 3.6 wt.% Ti; diisobutyl phthalate (DIBP) was used as an internal donor with a purity degree of 99.99% from Sudchemie, Germany. As a co-catalyst, triethylaluminium (TEAL) with a purity of 98%, obtained from Merck, Germany, was used as well as cyclohexyl methyl dimethoxy silane, the latter of which had a purity of 99.9%. Hydrogen and nitrogen gases obtained by Lynde were used. Finally, the propylene used was obtained from Airgas with a purity of 99.999%.

For the synthesis of polypropylene, the methodology proposed in [1,12] was followed, which suggests obtaining polypropylene with the help of a Ziegler–Natta catalyst [13,14,15,16,17,18]. The process starts with the use of a fluidized bed reactor which is purged with nitrogen; after this, the system is fed with hydrogen and propylene, which provide fluidization in addition to absorbing heat from the reaction. A Ziegler-Nata catalyst, TEAL, and a selectivity control agent were also incorporated along with nitrogen. The quantities are shown in Table 2. The process was carried out at 70 °C and 27 bar pressure in a discontinuous condition; the reactor product gases were captured with a compressor and then taken to a heat exchanger, where they were cooled and recirculated to the system. To obtain virgin resin, the method proposed by [19,20,21,22,23,24,25,26,27,28,29,30,31,32,33,34] was followed. The resin obtained from the reactor was taken to a purge tower fed with nitrogen to eliminate the hydrocarbons leaving the system.

### 2.2. Gas Chromatography Analysis with Selective Mass Detector (GC-FID)

The data obtained to express the quantification of furan in the green ethylene samples were obtained with the aid of a gas chromatograph (Agilent 7890B), with a front and rear injector of (250 °C, 7.88 psi, 33 mL min^−1^) and (250 °C, 11.73 psi, 13 mL min^−1^), respectively. The volume used ranged from 0.25 to 1.0 mL. The oven was started at 40 °C × 3 min, which increased to 60 °C–10 °C min^−1^ × 4 min and then increased to 170 °C at 35 °C min^−1^.

### 2.3. Melt Flow Index—MFI

The flow index was determined using the methodology proposed by [12], in which a Tinius Olsen MP1200 plastometer was used. The temperature inside the equipment was 230 °C, and a 2.16 kg piston was used to displace the molten material.

### 2.4. Thermogravimetric Analysis—TGA

For the development of the thermogravimetric analysis (TGA), a TGA Q500 thermal analyzer was used to achieve better results and continued with the methodology established by [12], where they proposed a heating rate of 10 °C min^−1^ from 40 to 800 °C in an air atmosphere (50 cm^3^ min^−1^). Therefore, the equipment was calibrated for temperature and weight via standard methods.

### 2.5. Analysis of Mechanical Properties

#### 2.5.1. Injection Molding

Several authors propose that materials obtained from thermoplastics should be processed with a twin-screw extruder and then subjected to an injection molding process [32]. These composites can also be molded via compression molding, among other techniques. For the present study, the samples were obtained via injection molding; to achieve better results when molding the samples, variables such as temperature and pressure were controlled, keeping the former at 50 °C and having strict control over the latter.

#### 2.5.2. Test Specimen Preparation

ASTM standards were followed to prepare the specimens. Two molds were used to design the tensile (ASTMD638) and flexural (ASTMD790) test samples to be stored for 48 h at the same temperature as the surrounding environment and then used for each of their respective tests.

#### 2.5.3. Tensile Test

To verify the resistance of thermoplastic composites to axial tensile forces and to determine the ability of the composite to elongate before cracking and failure, a tensile strength test was performed. The tensile test was performed according to the ASTMMD638 standard on a computerized H50KL universal testing machine (TiniusOlsen). The samples were clamped at both ends and then subjected to uniaxial tensile force. Since the samples were obtained via injection molding, they were classified as TYPE 1. The dimensions of the pieces to be studied were followed according to the methodology proposed by Suraj and Sanyay, 2022 [32], where the value of the gauge length (G) was 50 mm, the width of the narrow section (W) was 12.7 mm, and the thickness (T) was 3.4 mm.

#### 2.5.4. Flexural Test

Flexural strength is considered to be the capacity of a composite to withstand the bending force to which it is subjected, which is applied transversally to the shaft. For the development of this research, this resistance was evaluated by taking into account the ASTMD790 standard and using the H50KL universal testing machine (TiniusOlsen).

#### 2.5.5. Impact Test

Using the model IT504 pendulum impact tester (TiniusOlsen), the Izod impact test according to ASTMD256 was performed. Each specimen is assigned a 45°, 2.5 mm deep AV notch. One end of the notched specimen was fixed using a cantilevered vice.

## 3. Results

### 3.1. Effects of Furan on the Polymerization Process of Green Random Copolymer Rat

Figure 1 shows that the presence of furan was inversely proportional to the productivity of the Ziegler–Natta catalyst since the higher the furan concentration (ppm), the lower the catalyst’s effectiveness. In the absence of furan, the productivity of the ZN catalyst was 47 MT/kg. The average concentration of 6, 12, and 25 ppm of furan in ethylene affected productivity by 10, 20, and 41%, respectively.

The standard error for each of the variables was calculated based on the central limit theorem (see Equations (1) and (2)).
(1)$$Typical\;error=\frac{\sigma }{\sqrt{n}}$$
(2)$$\sigma =\sqrt{\frac{\sum_{i=1}^{N}{\left(x_{i}-\bar{x}\right)}^{2}}{N}}$$

It is essential to point out that all the groups studied presented significant differences (*p* < 0.05) when varying the concentration of furan from one group to another. This result is because furan acted as an inhibitor of the reaction since it reacted with the active center of titanium, preventing the propylene from polymerizing in the vibrant center of titanium. It should be noted that the inhibitory behavior of impurities on the productivity of the Ziegler–Natta catalyst has been previously reported [1,3,12,35,36,37,38,39,40], showing that the presence of impurities such as H_2_S, oxygenated compounds, and thiol, among others, affects the efficiency in the production of polypropylene on an industrial scale.

#### Proposed Mechanism of Action of Furan on the ZN Catalyst

Figure 2 shows the mechanism of furan formation in corn residues and other natural products. As mentioned in this study, two of the sources of green ethylene are corn and starch, which have glucose in their matrix. From this fact, Figure 2 shows the reaction mechanism for furan formation while obtaining ethylene from the fermentation of corn and starch. Figure 2 indicates that the furan formation process begins with four intramolecular rearrangements in three equilibria of the glucose molecule. The first structure moves an electron pair (double bond) from the carbonyl group to the alpha–beta carbon region to form an alpha–beta carbon–carbon double bond. In the second structure of Figure 2, the electronic pair transfers to form the carbonyl bond (C=O) on the beta carbon. The third equilibrium is the transfer of the electron pair to form a carbon–carbon beta–gamma double bond. Molecules 2 and 4 of the three proposed equilibriums undergo successive dehydration, losing hydroxyl groups and hydrogen atoms, with which the conditions for closing the ring between the beta–carbon carbonyl group and the epsilon carbon are obtained. In this way, the furan ring is thus formed. Molecule 2 additionally forms an equilibrium in which two electronic pairs are transferred from the carbonyl groups of the alpha and beta carbons to form alkene bonds between the alpha and beta and gamma and delta carbons; here, in the same way, dehydration occurs until the furan ring is formed, with the difference that in this way, a molecule of glycolic acid first detaches from the structure before starting the furan ring.

The furan originated in the pyrolysis of natural residues, which traces of unfermented glucose can generate. These glucose molecules can undergo a dehydration process in the first two steps of the mechanism presented in Figure 3, causing the characteristic and propitious condition at carbon 1,4 for the formation of the furanoid ring; later, this ring undergoes further dehydration, followed by dehydrogenation and the removal of the carbonyl group that combines with the ring, leaving hydrogen to form formaldehyde. This mechanism is because the pyrolysis conditions destabilize the glucose molecule, which decomposes to create more stable species under such conditions, such as furan.

In the mechanism proposed in Figure 4, furan competes with propylene monomer for the Ti-active site. First, a π complex is formed by coordinating the furan with the Ti of the TiCl_4_/MgCl_2_ complex. The furan-Ti interaction is carried out through the interaction of the electropositive Ti with the lone pair of electrons of the oxygen atom of the furan heterocyclic structure. This interaction is proposed to have a higher energy gain than that of a π complex in Ti-propylene. Therefore, the furan-Ti reaction predominates. In other investigations, it has been shown that to synthesize this family of polymers, there is a propylene insertion barrier that varies between 6 and 12 kcal mol^−1^ because the probability of the occurrence of the propylene insertion reaction in Ti is supported by thermodynamics since a favoring of approximately 20 kcal mol^−1^ is observed. For the efficiency of the reaction to be affected, the growth of the PP chain will be affected when the active site of Ti reacts with inhibitors of different polarities. Since furan exhibits intermediate polarities to other poisons such as H_2_S, their energy values are expected to be within the ranges of H_2_S. As shown in Figure 4, furan interferes with the formation of propylene complexes and their insertion.

### 3.2. Effects of Furan on the TGA of the Random Copolymer

TGA determined the thermal degradation of the copolymer; the curves are shown in Figure 2. In these, it can be seen that as the concentration of furan increases, its thermal stability decreases. This is due to the inhibitory capacity of furan in the polymerization process [12] and the favoring of the formation of new functional groups as a consequence of incomplete polymerization. This generates an alteration in the behavior and structure of the copolymer at the micromolecular and macro-molecular levels. When evaluating thermal degradation, it is evident that samples PP0 and PP1 have similar behavior in terms of weight loss, which is 5% by weight at 390 °C. For PP3 and PP4, there is loss of 5% at 370 °C and 320 °C, respectively as shown in Figure 5.

### 3.3. Effects of Furan on the MFI of the Green Random Copolymer

Figure 6 shows furan’s effect on the copolymer’s flow rate. Significant differences (*p* < 0.05) were observed when varying the furan concentration, observing a proportional relationship between the concentration of furan in the copolymer and the MFI index of the samples PP1, PP2, and PP3. The MFI of the copolymer without furan was 20 and increased to approximately 21, 23, and 27 g/10 min due to 6, 12, and 25 ppm furan, respectively. This corroborates what was observed in Figure 1, which shows that furan has an adverse action on the catalytic activity of ZN and, therefore, on the polymerization reaction. The chemical structure of furan may allow its oxygen atom in its heterocyclic ring to react with the active center of the titanium of the ZN catalyst to form a new stable complex that prevents the growth of the chain length of the obtained copolymer [1,12,22,29,30], its increase in fluidity, and the decrease in the molecular weight of the polymer. This can be seen in Figure 7, where the rise in furan concentration increases the MFI and decreases its molecular weight. This inhibitory behavior caused by polluting chemicals in the production of resins of industrial interest has been previously reported (Hernandez et al., 2022) and demonstrated that compounds such as Arsenia in concentrations of 0.001 to 4.32 ppm affect the MFI, and consequently, the molecular weight of the polymer.

### 3.4. Effects of Furan on the Mechanical Properties (Tensile, Flexural, and Impact) of the Random Green Copolymer

The influence of the presence of furan on the mechanical properties of the copolymer is shown in Figure 8, Figure 9 and Figure 10. When evaluating the mechanical properties of the obtained product, a notable difference is observed in the PP0 copolymer compared to PP1, PP2, and PP3, which allows us to affirm that the presence of furan can directly affect these properties of the copolymer since as the concentration of the component increases, the values of tension, bending, and impact decrease. PP0 presented average bending, tensile, and Izod values of 211,160 psi, 411,833 psi, and 11.6 ft-Lb*in, respectively. Increasing furan concentrations by an average of 6, 12, and 25 ppm caused flex decreases of 1, 11, and 18%, respectively. The tensile decreases were 4, 13, and 18%, respectively. The Izod impact trend was also inversely proportional to the furan concentration, showing percentage decreases of 9, 18, and 22%. This variation in the data is mainly due to the changes generated at the structural level that occur in the copolymer, a product of the formation of new compounds with different functional groups, in addition to incomplete polymerization and the presence of an oxygen atom in the furan, directly influence the mechanical properties of the copolymer. This behavior is associated with low flow rates and molecular weight distribution, which directly affect the mechanical properties of polypropylene [24].

## 4. Conclusions

In the present investigation, an analysis of a random green copolymer was carried out to determine the influence of the presence of furan on the mechanical properties, MFI, productivity, and TGA of the copolymer. The results showed an evident effect of furan on the random copolymer polymerization process, inhibiting the Ziegler–Natta catalyst’s capacity in this process, which generated losses in the productivity of this catalyst. A notable decrease in the thermal capacity of copolymer was evidenced, and the results were supported by the TGA levels obtained, as well as an increase in the MFI levels, which may have been associated with a decrease in the molecular weight values due to the degradation of the chemical structure of copolymer. Finally, a remarkable dominance of furan on the mechanical properties of the random copolymer was observed due to the changes generated in the levels of MFI, thermal degradation, and loss of the productivity of the catalyst.

## Figures and Tables

**Figure 1 polymers-15-02264-f001:**
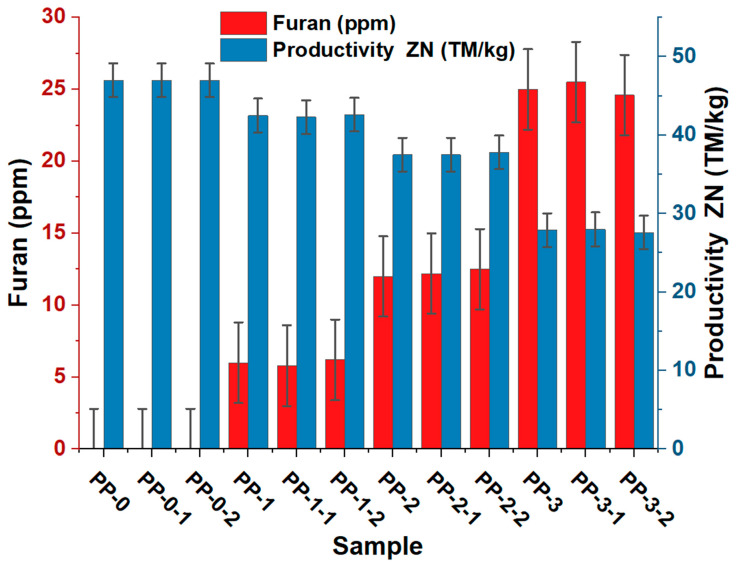
Productivity lost in the polymerization process of the green random copolymer.

**Figure 2 polymers-15-02264-f002:**
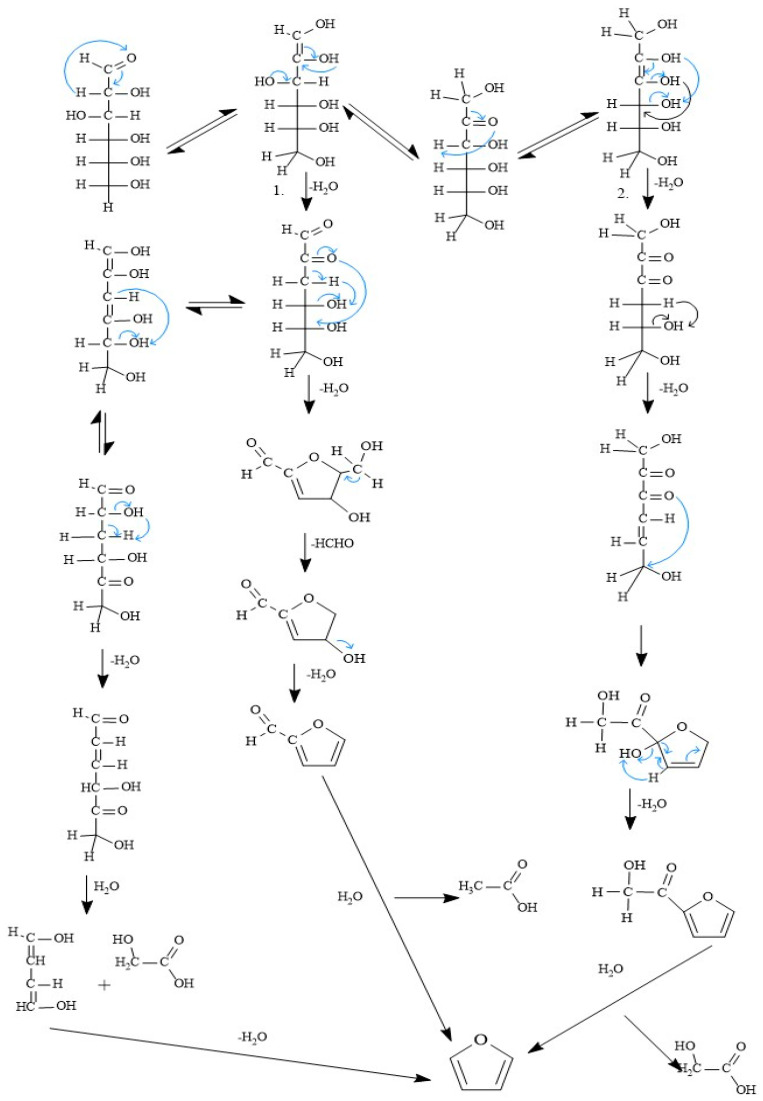
Proposal for the mechanism of furan biogenesis.

**Figure 3 polymers-15-02264-f003:**
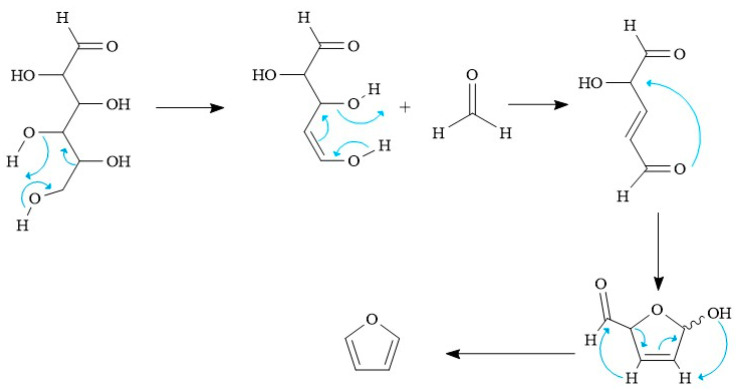
Proposal of the mechanism of furan formation during the pyrolysis of natural residues.

**Figure 4 polymers-15-02264-f004:**
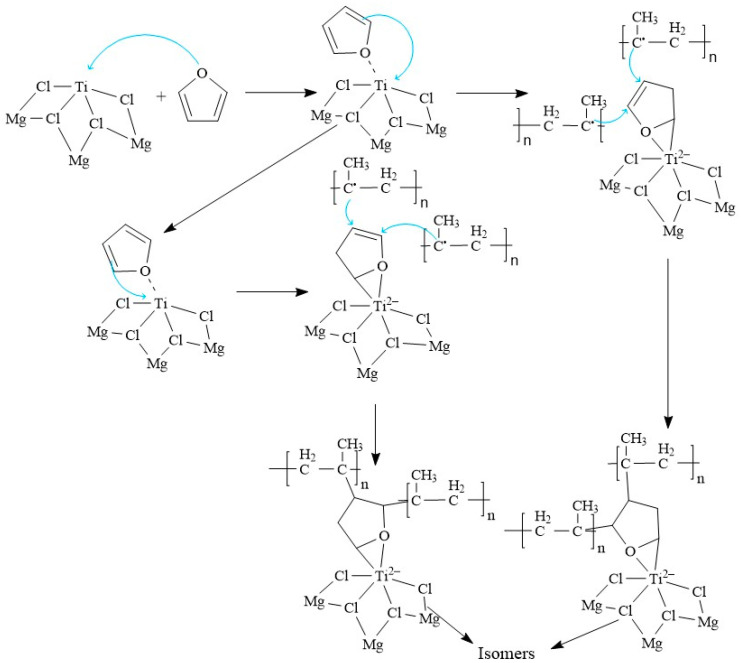
Proposal of the reaction mechanism between furan and the Ziegler–Natta catalyst.

**Figure 5 polymers-15-02264-f005:**
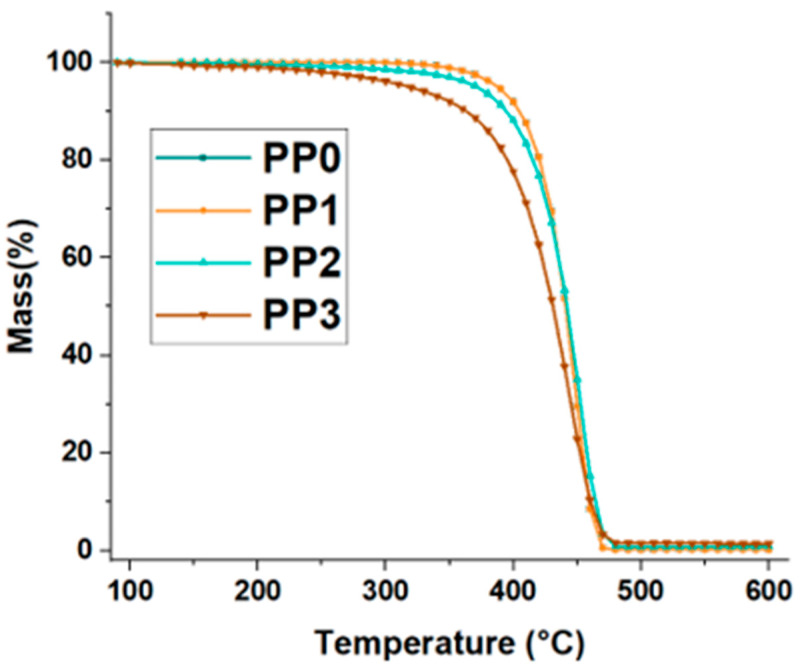
Effects of furan on thermal degradation of the random copolymer.

**Figure 6 polymers-15-02264-f006:**
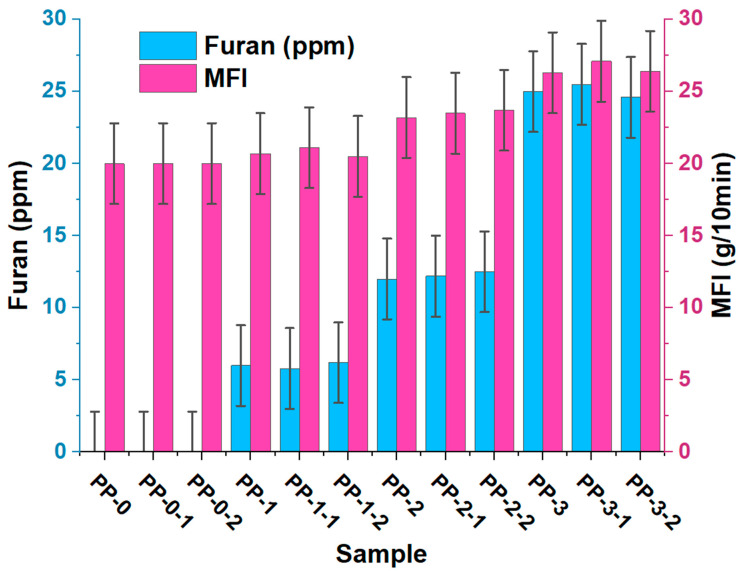
Effects of furan on the MFI of the random copolymer.

**Figure 7 polymers-15-02264-f007:**
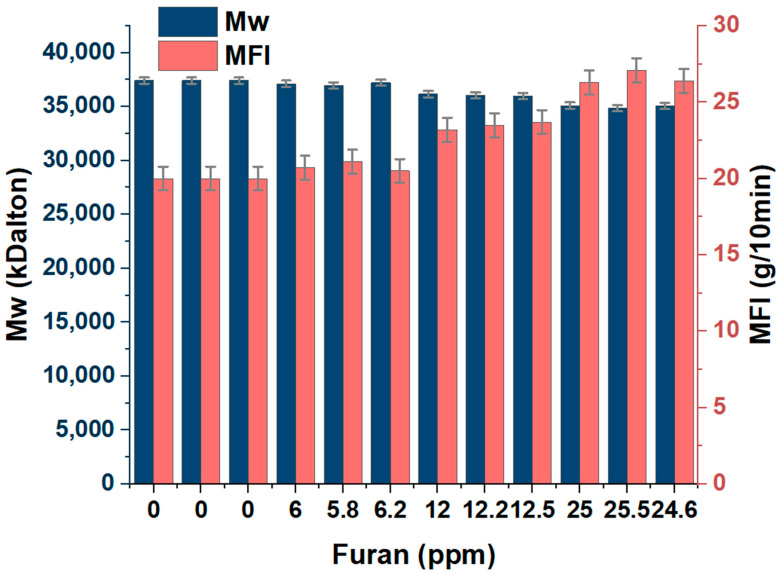
Effects of furan on the Mw and MFI of the random copolymer.

**Figure 8 polymers-15-02264-f008:**
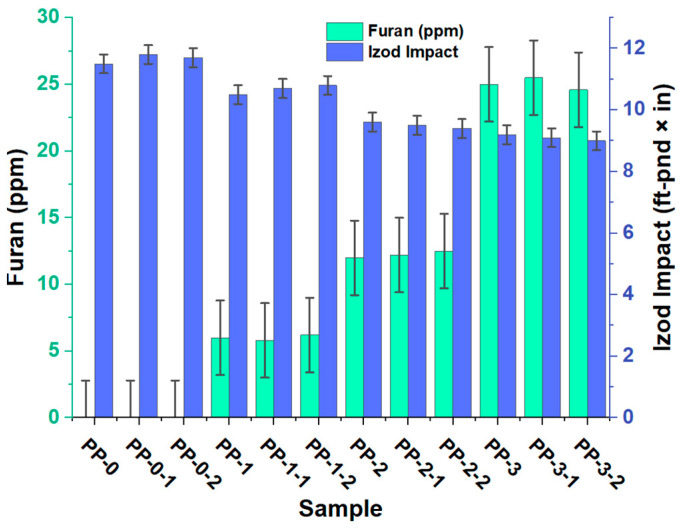
Effects of furan on the impact of the random copolymer.

**Figure 9 polymers-15-02264-f009:**
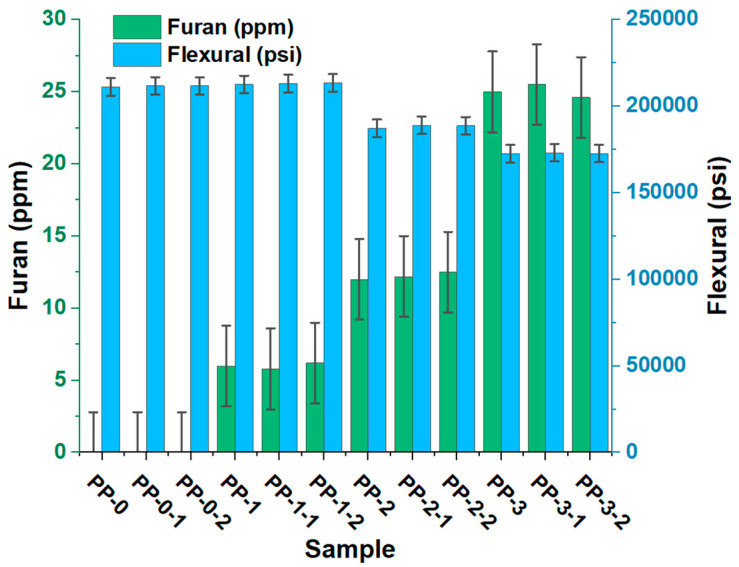
Effects of furan on the flexural of the random copolymer.

**Figure 10 polymers-15-02264-f010:**
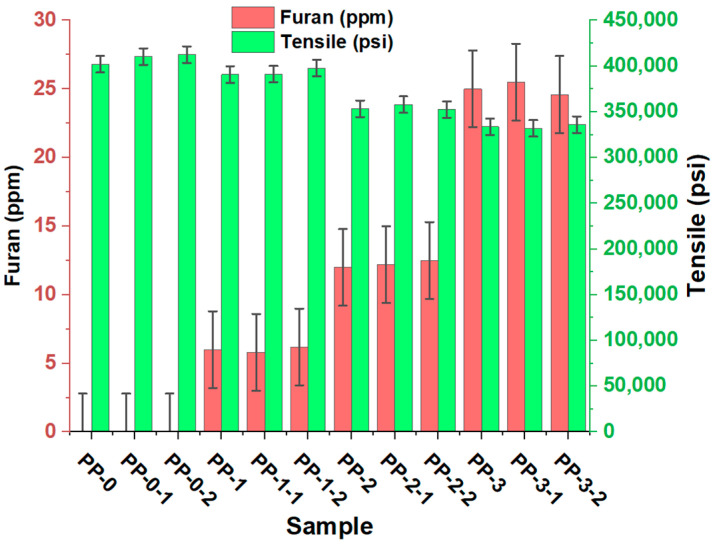
Effects of furan on the tensile of the random copolyme.

**Table 1 polymers-15-02264-t001:** List of reagents for obtaining polypropylene.

Materials	Supplier Used	Purity
Diisobutyl phthalate (DIBP)	(In-house donor) Sudchemie, Germany	99.99%
Triethylaluminium	(Co-catalyst)Merck, Germany	98%
Cyclohexyl methyl dimethoxysilane (CMDS)	(External donor) Merck, Germany	99.9%
Hydrogen	Lynde	99.999%
Nitrogen	Lynde	99.999%
Ethylene	Airgas	99.999%
Propylene	Airgas	99.999%

**Table 2 polymers-15-02264-t002:** Identification of samples and sampling points.

Materials	Run											
	PP0	PP0-1	PP0-2	PP1	PP1-1	PP1-2	PP2	PP2-1	PP2-2	PP3	PP3-1	PP3-2
**Catalyst, kg/h**	5	5	5	5	5	5	5	5	5	5	5	5
**Propylene TM/h**	1.2	1.2	1.2	1.2	1.2	1.2	1.2	1.2	1.2	1.2	1.2	1.2
**Green ethylene**	0.6	0.6	0.6	0.6	0.6	0.6	0.6	0.6	0.6	0.6	0.6	0.6
**TEAL, kg/h**	0.25	0.25	0.25	0.25	0.25	0.25	0.25	0.25	0.25	0.25	0.25	0.25
**Hydrogen, g/h**	30	30	30	30	30	30	30	30	30	30	30	30
**Furan (ppm)**	0	0	0	6	5.8	6.2	12	12.2	12.5	25	25.5	24.6
**Selectivity control agent, mol/h**	1	1	1	1	1	1	1	1	1	1	1	1
**T, °C**	70	70	70	70	70	70	70	70	70	70	70	70
**Pressure, bar**	27	27	27	27	27	27	27	27	27	27	27	27

## Data Availability

The data presented in this study are available on request from the corresponding author.

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
