# Peer review of "Furan as Impurity in Green Ethylene and Its Effects on the Productivity of Random Ethylene–Propylene Copolymer Synthesis and Its Thermal and Mechanical Properties"

_polymers, 2023, doi:10.3390/polym15102264_

Round 1
Reviewer 1 Report
This manuscript describes the influence of furan impurity in polyolefin synthesis. The following needs to be addressed for this work to be considered for publication in Polymers:
1. The motivation for the authors to investigate furan, specifically, in this work is unclear to readers. Why is furan being chosen as the candidate? Is furan the most significant side-product of the production of green ethylene? Is it the highest amount in the starting material? How is it being produced? What is the mechanism? The authors should also include structures of these chemicals and related transformation.
2. It is unsurprising to the reviewer that heterocyclic compounds such as furan will react with the Ziegler Natta catalyst and affect the polymerization. So the conclusion of this manuscript is not strong enough. Can the authors expand the scope and investigate other types of impurities? Any suggestions to purify green ethylene from these heterocyclic compounds?
Author Response
- The motivation for the authors to investigate furan, specifically, in this work is unclear to readers. Why is furan being chosen as the candidate?
R/ Thank you for reviewing this research. In our line of research, we study the efficiencies of polypropylene production reactions using Ziegler Natta catalysts. In this sense, in other investigations, we have evaluated different inhibitors such as H2S, COS, methyl mercaptan, acetylene, etc. The long-term goal in our future research is to have mapped all the inhibitors that affect the productivity of the Ziegler Natta catalyst. Through other investigations, we have shown that Furan is obtained in trace levels when green ethylene is produced. So since this furan is found in trace levels in green ethylene, It is super important to know its impact on the synthesis of the copolymer.
2- Is furan the most significant side-product of the production of green ethylene?
R/ Thank you for reviewing this research. Furan is an impurity present in green ethylene.
3-Is it the highest amount in the starting material?
R/No, it's just an impurity
4-How is it being produced?
R/This information is not known experimentally. We have now proposed a theoretical route of the formation
5-What is the mechanism?
R/Now we have proposed it.
6-The authors should also include structures of these chemicals and related transformation.
R/Thank you for reviewing this research. Done..
7-It is unsurprising to the reviewer that heterocyclic compounds such as furan will react with the Ziegler Natta catalyst and affect the polymerization. So the conclusion of this manuscript is not strong enough. Can the authors expand the scope and investigate other types of impurities? Any suggestions to purify green ethylene from these heterocyclic compounds?
R/In our line of research, we have studied several inhibitors that affect polypropylene synthesis. Still, we need to broaden the spectrum of molecules that affect the catalyst's efficiency. Also, know the ranges of concentrations in which these inhibitors cause damage to the catalyst. This last part is more complex and takes a long time, but we keep it in mind. On this occasion, we have studied furan, mainly because it is an impurity present in the raw material and at an industrial level in large productions it has been considered that when this molecule is present as an impurity, the efficiency of the catalyst and the properties of the PP are affected. So in this particular investigation, it is demonstrated how its presence in different concentrations affects the synthesis. In another investigation that we are developing, we are modifying zeolites to remove residues of this furan in ethylene.
Reviewer 2 Report
The proposed study is devoted to the invesitigation of the influence of furan on the catalytic activity of catalysts of polymerization and properties of the obtained polymers. The topic of Ziegler Natta catalysts are of high interest for the readers. However, this manuscript has some phenomenological and technical moments. The following questions should be addressed:
1) the novelty of the work shoudl be highlighted in introduction and abstract
2) difference from recently published paper should be specified (https://doi.org/10.1007/s10118-018-2092-0)
3) there is many misprints with word "furan", incorrect versions are in figures (furane, furano) and in the text
4) please, add erroro bars to fig. 1, 3-7
Author Response
The proposed study is devoted to the invesitigation of the influence of furan on the catalytic activity of catalysts of polymerization and properties of the obtained polymers. The topic of Ziegler Natta catalysts are of high interest for the readers. However, this manuscript has some phenomenological and technical moments. The following questions should be addressed:
We have made improvements to the process. We have proposed some mechanisms to understand the origin of the furan and later to know its reaction with the catalyst.
1) the novelty of the work shoudl be highlighted in introduction and abstract
R/ Thank you for reviewing this research. We have written the information.
2) difference from recently published paper should be specified (https://doi.org/10.1007/s10118-018-2092-0)
R/ In our line of research, we have studied several inhibitors that affect polypropylene synthesis. Still, we need to broaden the spectrum of molecules that affect the catalyst's efficiency. Also, know the ranges of concentrations in which these inhibitors cause damage to the catalyst. This last part is more complex and takes a long time, but we keep it in mind. On this occasion, we have studied furan, mainly because it is an impurity present in the raw material and at an industrial level in large productions it has been considered that when this molecule is present as an impurity, the efficiency of the catalyst and the properties of the PP are affected. So in this particular investigation, it is demonstrated how its presence in different concentrations affects the synthesis. In another investigation that we are developing, we are modifying zeolites to remove residues of this furan in ethylene.
3) there is many misprints with word "furan", incorrect versions are in figures (furane, furano) and in the text
R/Thank you for reviewing this research. We have made the suggested corrections.
4) please, add erroro bars to fig. 1, 3-7
R/Thank you for reviewing this research. We have made the suggested corrections.
Round 2
Reviewer 1 Report
This manuscript can now be accepted in Polymers.